# Detecting Banana Plantations in the Wet Tropics, Australia, Using Aerial Photography and U-Net

**Andrew Clark *** and **Joel McKechnie**

Remote Sensing Centre, Queensland Department of Environment and Science, GPO Box 2454, Brisbane,
QLD 4001, Australia; joel.mckechnie@hotmail.com

\* Correspondence: andrew.clark@des.qld.gov.au; Tel.: +61-7-31705669

**Abstract:** Bananas are the world's most popular fruit and an important staple food source. Recent outbreaks of Panama TR4 disease are threatening the global banana industry, which is worth an estimated \$8 billion. Current methods to map land uses are time- and resource-intensive and result in delays in the timely release of data. We have used existing land use mapping to train a U-Net neural network to detect banana plantations in the Wet Tropics of Queensland, Australia, using high-resolution aerial photography. Accuracy assessments, based on a stratified random sample of points, revealed the classification achieves a user's accuracy of 98% and a producer's accuracy of 96%. This is more accurate compared to existing (manual) methods, which achieved a user's and producer's accuracy of 86% and 92% respectively. Using a neural network is substantially more efficient than manual methods and can inform a more rapid respond to existing and new biosecurity threats. The method is robust and repeatable and has potential for mapping other commodities and land uses which is the focus of future work.

**Keywords:** convolutional neural network; U-Net; segmentation; deep learning; land use; banana plantation; Panama TR4; aerial photography

## 1. Introduction

### 1.1. Panama TR4

*Fusarium oxysporum* f. sp. *cubense* tropical race 4 (Foc TR4), is a soil-borne fungus that causes Panama TR4, a form of fusarium wilt that eventually kills infected banana plants [1,2]. Since the 1980s, Foc TR4 has been regarded as the most important biosecurity threat to the global banana industry, and an unparalleled botanical epidemic [2], persisting indefinitely in the soil with no effective control method. There is currently no suitable replacement variety for Cavendish that can meet the needs of the market [1,3,4]. The disease can spread anthropogenically and naturally through the transportation of infected plant material, soil, and water [5].

Bananas are the world's most popular fruit and an important staple food [6], with a global industry worth \$8 billion annually [7]. The potential impact of Panama TR4 is severe, because Cavendish accounts for approximately 47% of bananas produced globally, predominantly sourced from Asia, Latin America, and Africa [7].

Foc TR4 was first identified in Sumatra, Indonesia, in 1992 [8], and to date, has spread across several continents [1,9,10] including Australia; the Northern Territory in 1997 and the state of Queensland in 2015 [4,8]. The Queensland Government Department of Agriculture and Fisheries (DAF) initiated the Panama TR4 Program in response to the first detection in the Tully River catchment, within the Wet Tropics bioregion. The program successfully controlled and contained the impact of Panama TR4 within a section of the Tully River catchment, however, three additional plantations within this location

were infested in 2017, 2018, and 2020 [9,10]. At the time of the outbreak there was no accurate spatial dataset of all banana plantations [11]. The absence of this data jeopardized the banana industry and DAF's abilities to respond rapidly. Approximately 94% of the national banana supply is concentrated in the Wet Tropics, and is worth approximately AUD 480 million annually to the national economy [12]. Therefore, it is essential that the locations and extents of affected, and unaffected, banana plantations are monitored—particularly where vectors are likely to be transported, for example through erosion of contaminated soil, distribution and processing facilities, and machinery and equipment used across multiple plantations.

*1.2. Land Use Mapping*

The Queensland Government, currently through the Department of Environment and Science, has mapped land use and land use change throughout Queensland since 1999. The Queensland Land Use Mapping Program (QLUMP) maps land use in accordance with the Australian Land Use and Management (ALUM) classification [13]. In 2015, when Panama TR4 was first detected in the Wet Tropics, banana plantations had not been specifically classified, as they did not explicitly appear in the ALUM classification. For DAF and the banana industry to manage the Panama TR4 infestation, Système Pour l'Observation de la Terre (SPOT) 6 imagery was acquired over the Wet Tropics, for QLUMP to manually digitize the extent of banana plantations.

Timely land use mapping is fundamental for responding to biosecurity incidents, and for other applications such as natural disaster response, natural resource management and environmental monitoring [14]. Advances in big data and imagery availability have created an opportunity to develop methods to automatically and efficiently classify land use features over large geographical areas to allow for higher spatial and temporal resolutions and a more detailed classification for commodity level observations.

Using high-resolution imagery, many different land uses can be identified with human vision, including banana plantations. This is a result of human operators combining a number of image properties including colors, textures, pixel proximity, geometric attributes, and contextual information such as related built infrastructure [15]. Spectral information alone cannot successfully distinguish land use features as some land uses can appear spectrally similar [16] and are usually restricted to a single sensor without cross calibration. Using ancillary datasets and decision trees to derive land use is not always an accurate representation of what is on the ground [14].

The greater availability of high-resolution imagery introduces more complexity into the data, requiring more computing power to process the imagery and more detailed classifications. The integration of textural properties through object-based segmentation techniques have significantly improved the classification results for remote sensing applications [15]. For land use mapping, it has been found that using spatial as well as spectral information outperforms pixel-based classifications [17]. However, object-based image analysis approaches still require human input [16,18] and these complex workflows tend to be just as time- and resource-intensive as entirely drawing the land use features manually [19–21].

*1.3. Deep-Learning Classifications*

Neural networks have been around for many decades (see review by Schmidhuber [22]). However, only since the recent advancements in GPU technology have they been able to be trained with large amounts of data in a reasonable amount of time [23]. Neural networks simulate the processes of the human brain—interconnected neurons which process incoming information [24]. The solution is obtained by nonalgorithmic and unstructured methods, and by the adjustment of weights connecting the neurons in the network [25]. They can adaptively simulate complex and non-linear patterns [24,26] such as those found in high-resolution aerial photography.

Deep learning methods are based on neural networks [22]. These networks consist of many layers, which can transform images into categories through learning of high-level features [27]. Convolutional

Neural Networks (CNN) are situated at the fringe between machine learning and computer vision, combining the power of deep learning with contextual image analysis. CNNs have been used in applications such as number-plate reading, facial recognition, and aerial image classification [28,29]. CNNs have gained momentum for image classification since the AlexNet architecture won the ImageNet contest by a wide margin in 2012 [30].

Mnih [31] and Romero et al. [32] found that deep CNNs outperform shallow CNNs (with fewer hidden layers), Support Vector Machines (SVM), Kernel-based Principal Component Analysis (KPCA), and spectral classifications for land use classification in aerial photography, multispectral and hyperspectral imagery. As CNN classifications integrate spatial as well as spectral information, they achieve higher accuracy compared to SVM and Random Forest (RF) classifications [23].

CNNs evaluate large amounts of contextual information over multiple scales that can result in classifications at a lower resolution than the original image. To overcome this, the review article by Ma et al. [23] suggests using U-Net [33] or an ensemble of models trained with different variables or different architectures. U-Net was originally developed for image segmentation problems in biomedical imaging [33] and has been adopted for use with optical earth observation data with overall accuracies exceeding 90% [34–36]. Issues occur with U-Net at the edges of inference areas and vanishing gradient problems where the network becomes difficult to train and has insufficient learning [37]. To overcome this, Sun et al. [37] suggests using an ensemble model approach to counter the edge effects and the use of concatenation operations and activation functions such as rectified linear units (ReLU) to reduce the vanishing gradient problem.

### 1.4. Automated Land Use Mapping

Most studies that have used deep learning to automatically map land use features have a constrained geographical extent and are limited to a standard set of training images, for example the University of California Merced Land Use Dataset [38] and Banja-Luka [39]. Although this is advantageous in benchmarking different methodologies, no studies have operationalized the applications for real-world land use mapping over a large geographical area [23,40].

Previous work in Queensland has focused on land cover. Pringle et al. [41] developed a time-series based method to operationally map summer and winter crops, and Flood et al. [42] used U-Net to map land cover, specifically the extent of woody vegetation cover, to a resolution of 1 m. However, there is an absence of studies that classify land use in Australia using earth observation data with a resolution of less than 10 m which is required for detailed land use mapping [43,44] and the resolution in which CNNs have the most success [23].

The aim of this study is to demonstrate that using a convolutional neural network and high-resolution imagery (<1 m) to automate land use mapping is more rapid and accurate than existing manual methods. This would be of benefit to the on-going response to Panama TR4, future biosecurity incidents, and other events requiring a rapid response (e.g., natural disasters). Additionally, the improved land use data would better inform natural resource planning and monitoring, biodiversity conservation, and the monitoring and modelling of the effects of land management practices on water quality.

## 2. Materials and Methods

### 2.1. Study Area

The location of this study is within the Wet Tropics and Atherton Tablelands, located approximately 1200 km northwest of Brisbane, Australia (Figure 1). The region of 2.7 million hectares, includes the Wet Tropics World Heritage Area, and is adjacent to the Great Barrier Reef World Heritage Area.

In 2015, QLUMP reported the major secondary land uses within the project area to be: Nature Conservation (37.5%); Grazing (31.52%); Other Minimal Use (8.2%); and Cropping (7.2%). There were 14,533 hectares (0.65%) of banana plantations mapped.

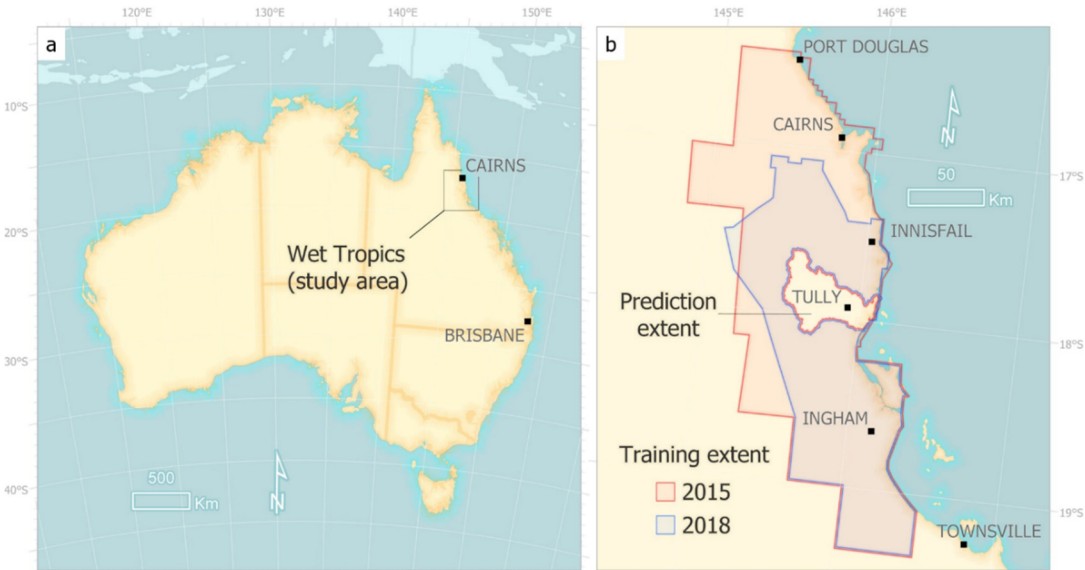

**Figure 1.** The study area (**a**) and extents of training (**b**). Note the Tully River catchment was excluded from the training and reserved as the extent of the final classification.

## 2.2. Remote Sensing Imagery

Two aerial imagery captures were used for this study (Figure 1). The 2015 data were acquired between 17th July and 14th October 2015, and the 2018 data were acquired between 1st and 27th August 2018, by AEROmetrex. At the time of this study, the full extent of the 2018 Wet Tropics imagery capture was not available so the training was restricted to the middle and southern sections of the region.

The data were captured with a fixed-wing mounted three-band true-color A3 Edge camera, at a spatial resolution of 25 cm and 20 cm for 2015 and 2018 respectively. The data were provided orthorectified by AEROmetrex based on a digital terrain model from LiDAR and stereo aerial imagery. The quality of the imagery is not consistent across the study area with some blurred areas and some discoloration. These artefacts are likely a result of post-processing of the imagery and appear to be located along tile boundaries where stitching the tiles and color balancing was not perfect. Unfortunately, the metadata supplied with the data does not list specific processing details. However the imagery is the best data available for the project area at a resolution suitable for this type of application. The Queensland Government has a large archive of aerial photography and it is likely these data, along with future captures will contain similar artefacts and any model developed will need to be robust enough to account for these inconsistencies.

## 2.3. Project Hardware and Software

Scripts and tools were written using the Python programming language. A combination of NumPy and Geospatial Data Abstraction Library (GDAL) were used to process the imagery and training data and convert them into multi-dimensional arrays, the format required for machine learning data processing.

A combination of Python, Nvidia's CUDA [45], CUDA deep neural network library (cuDNN) [46], Keras [47], and Tensorflow [48] formed the basis for the deep learning part of the project. Because of the volume of data and the number of iterations required to train a model, efficient processing of the data is required.

The Queensland Government's high performance computing (HPC) infrastructure consists of 2256 threads, 8.8 TB of memory and eight Nvidia Tesla V100 GPUs, used to process the training data, train the U-Net model, and to create the model inference (resulting area of banana plantations classified by the model once trained).

## 2.4. Existing Land Use Data Set

As described in Section 1.2, state-wide land use information is mapped by QLUMP. Land use is mapped to a national standard, according to the ALUM Classification—which has a three-tiered hierarchical structure broadly structured by the potential degree of modification from essentially native land cover [13]. The (six) primary and (32) secondary classes relate to land use, and (159) tertiary classes include commodity and land management practice information (e.g., "Tree fruits" as demonstrated in Figure 2. While tertiary-level information is particularly valuable for many applications, including biosecurity response, it has historically been expensive and impractical to collect, and as a result not consistently recorded.

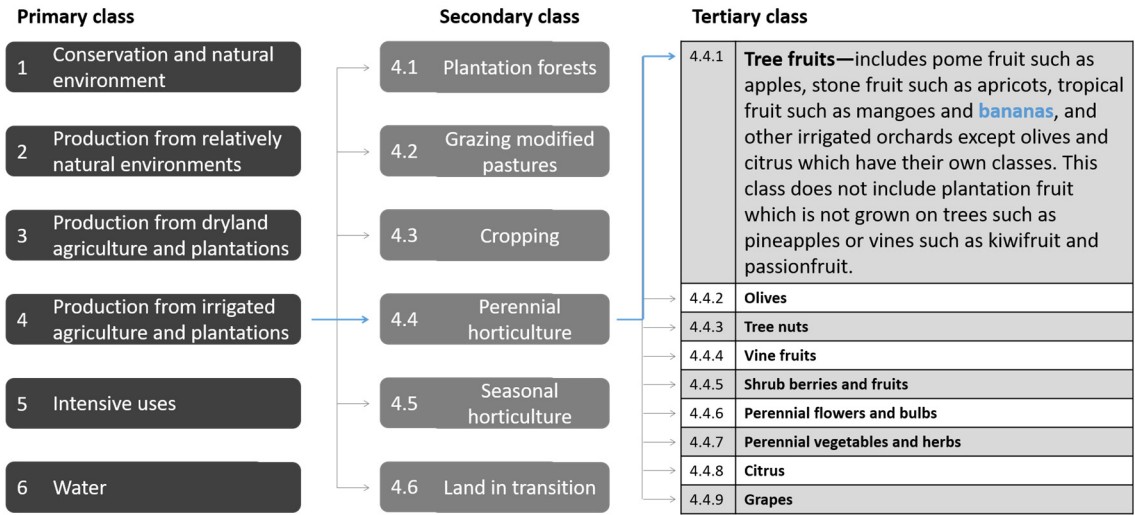

**Figure 2.** This diagram shows the three-tiered hierarchical structure of the Australian Land Use and Management (ALUM) classification and an extract demonstrating bananas as a commodity within the "Tree fruits" tertiary class, "Perennial horticulture" secondary class, and "Production from irrigated agriculture and plantations" primary class.

The QLUMP methodology has been an accurate, reliable, and cost-effective option since the late 90s—making use of available technology, data, and imagery. Mapping is undertaken primarily at the desktop, combining imagery interpretation and ancillary data to derive land use products. These products are field validated, peer reviewed, and accuracy assessed prior to publishing.

Because of the large area of Queensland (1.85 million squared kilometers), it has not been feasible to update land use information across the entire state at once, therefore updates occur regionally, using natural resource management (NRM) region boundaries. As a result, the currency of data varies from region to region. The most recent data is 2017 (Fitzroy and Burnett Mary NRM regions) and the most dated is 2012 (South East Queensland NRM region). The Wet Tropics NRM region was last updated to 2015. Regional updates occur on an ad hoc basis, dependent on state government priorities—for example the most recent updates were in the Great Barrier Reef catchments to support the Paddock to Reef Monitoring, Modelling and Reporting Program, and the Reef 2050 Water Quality Improvement Plan. A user survey conducted by QLUMP in 2020 indicates that there is a growing need for more current, and higher resolution land use information. The current QLUMP methodology, while proven, requires an intensive amount of manual image interpretation and spatial data analysis. There is a need for a more automated methodology that enables faster publishing of land use information, and CNNs are a possible solution.

## 2.5. Training Data

The generation of training data was an iterative process (Figure 3). Initially a subset of the existing QLUMP data in the study area was edited to better represent the land use features within the

2015 imagery. This editing was required as the QLUMP data were compiled using lower resolution imagery (SPOT 6 with a resolution of 1.5 m) and mapped land use features at a scale of 1:50,000 (using a minimum mapping unit area of 2 ha and width of 50 m) [13,49]. From these data, image and corresponding mask chips were randomly generated for model training. An initial inference was produced, converted to a polygon using a prediction probability threshold of 50%, and edited to fix any areas of omission and commission errors. This resulted in a more accurate and detailed training dataset compared to the QLUMP data. The image and mask chips were then regenerated for additional training.

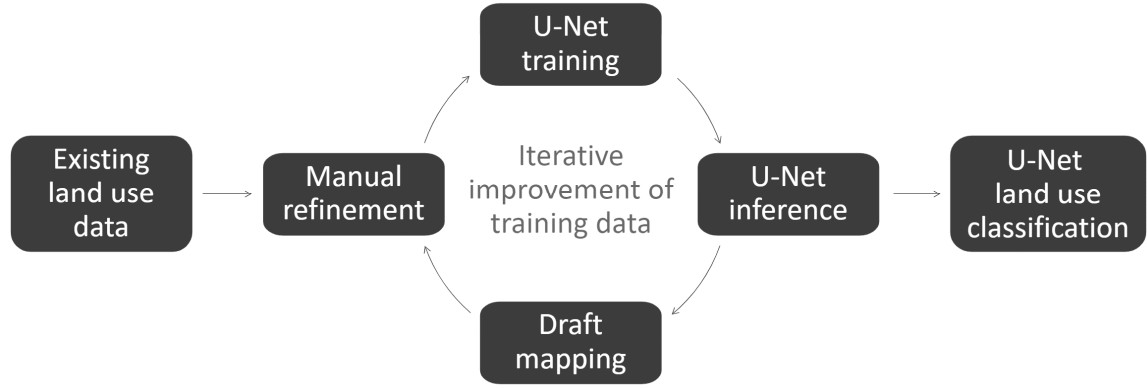

**Figure 3.** The training–inference–refinement iteration loop.

A total of 91,129 image chips with a size of 256 × 256 pixels were randomly generated from the 2015 and 2018 imagery. Of the total number of chips, 16,560 (18.2%) contained banana plantations, with the remainder located over a range of other land uses. No data were generated which intersected the Tully River catchment (Figure 1b) and this area was not included in the training stage of the study.

### 2.6. The U-Net Convolutional Neural Network

In this study, we aimed to classify every pixel in the image as banana or non-banana plantation through semantic segmentation using a convolutional neural network. The structure of the CNN follows the U-Net architecture [33] and is shown in Figure 4. It consists of two parts: (i) An encoding stage that downsamples the resolution of the input images; and, (ii) a decoding stage that upsamples and restores the images back to the original resolution. At each level of the encoding stages, two 3 × 3 convolution operations are applied using the rectified linear unit (ReLU) activation and a 2 × 2 max pooling operation to downsample the input images. The first level consists of the original satellite image and mask chips (256 pixels in width and height) where 64 3 × 3 filters are applied to each chip. At each subsequent level of the encoding side of the U-Net, the number of filters is doubled, doubling the number of bands of the images and the resolution halved until the bottom level where 1024 filters are applied to images 16 × 16 pixels in size.

The decoding stage also uses two 3 × 3 convolution operations but upsamples the data and concatenates the corresponding information in the encoding stage to double the resolution of the images, eventually restoring the original resolution of the input images in the final level. The final step is to conduct a 1 × 1 convolution using a sigmoid activation to produce a single band output probability classification with values ranging from 0 to 1. Values closer to 1 are more likely to be banana plantations. Using this configuration of the U-Net allows for the training of 31.4 million parameters overall.

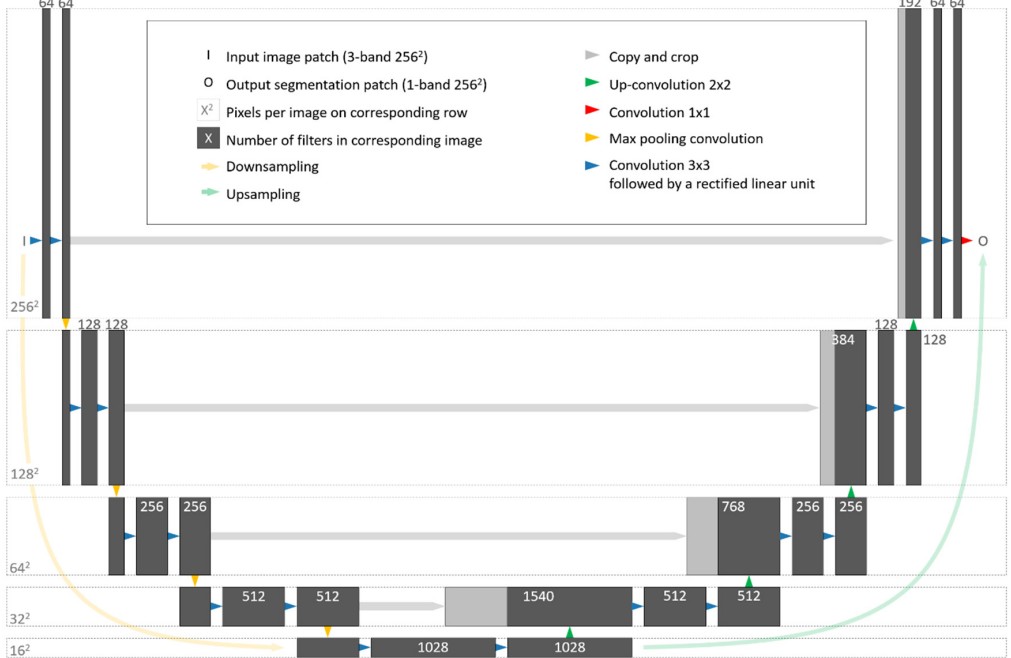

**Figure 4.** The U-Net architecture, starting with an input image patch (top-left) and ending with an output segmentation patch (top-right) [33].

### 2.7. U-Net Training

The purpose of the training stage is to allow the model to learn how to identify banana plantations. This is achieved by iterating over the training image and mask chips to determine their relevant color, texture, and context attributes [50]. As the images were captured over a range of image dates, subjected to color balancing and not corrected to surface reflectance, the training patches were randomly augmented by flipping, rotating, and changing the brightness of the image. This creates a more robust model for these image types [51,52].

A loss function of binary cross entropy and the Jaccard Index was used to judge the performance of the model while training. The Nesterov Adam optimizer [47] with an initial learning rate of $1 \times 10^{-5}$ was used and reduced to $1 \times 10^{-6}$ at epoch 47 as the model accuracy was no longer improving. The model was trained from scratch for a total of 50 epochs. One epoch represents one complete iteration over all training images and masks. The model took approximately 30 h to train on one Nvidia Tesla V100 GPU.

### 2.8. U-Net Inference

Sun et al. [37] found in previous studies that the edges of each image chip have a lower accuracy than the center region. To overcome this, a two-pass ensemble inference strategy was adopted. This was done by breaking the whole image into $256 \times 256$ image chips, iteratively applying the model to the original patch and three rotated versions of the patch and averaging the results. The second pass of the image was conducted, offset by 128 pixels. The result from the two passes are combined using a weighted average with pixels toward the center of the patch given a higher weight than the pixels toward the edge.

A prediction probability threshold of 90% was used to classify areas of banana plantations. Small features and gaps of 0.01 hectares or less were removed or filled and the data were converted into a polygon feature class with the edges smoothed to remove the square edges of individual pixels.

To increase the performance of the classification, the original aerial images were split up into overlapping tiles, allowing the inference to be conducted on all eight Nvidia Tesla V100 GPUs. It took approximately 12 h to run the model on the 2015 imagery.

*2.9. Accuracy Assessment*

Two independent assessment measures were conducted to assess the accuracy of the U-Net and QLUMP classifications of banana plantations in the Tully River catchment.

The first assessment was based on a stratified random sample of 9805 points using the ALUM Classification tertiary classes as the strata, following the method described in [53]. As the scale of the QLUMP data was different to the high-resolution imagery used to map land use classes, every point was visually inspected to ensure they were correctly classified as a banana plantation or other land use. If there were inconsistencies between the classification of a point and the imagery, the point was reclassified to the correct land use class. For example, if a banana plantation point fell on an area of fallow or a narrow road between the banana plantations, these points were reclassified as land-in-transition or road. The points were used to calculate the user's, producer's, and total accuracies for the U-Net and QLUMP classifications of banana plantations. In total, 701 out of 9805 points were located on banana plantations (7.15%).

The second measure of accuracy for both classifications was conducted using a similarity coefficient, the Jaccard Index defined in Equation (1). The benefit of using the Jaccard index for measuring the accuracy of the banana plantation classification is that it accounts for all the validation data excluded from the training of the U-Net model for the Tully River catchment. It is a similarity index and compares how well the validation and classification banana plantation locations match.

$$J(C, G) = \frac{C \cap G}{C \cup G} \tag{1}$$

where $C$ is the classification (U-Net or QLUMP) and $G$ is the validation data.

## 3. Results

*3.1. U-Net Training*

After 50 epochs, the model achieved a Jaccard Index of 0.961 and a loss of 0.01 (Figure 5). Because of the high-quality training data used, the model achieved a Jaccard Index of 0.8 after 4 epochs and 0.9 after 11 epochs.

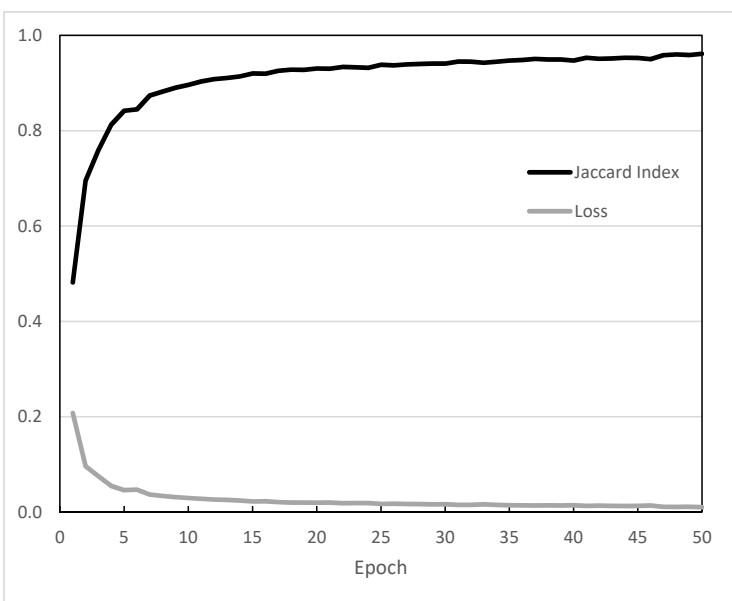

**Figure 5.** The Jaccard Index and loss function for each epoch. The number of epochs was restricted to 50. A marginal improvement may have been recorded if allowed to continue.

### 3.2. U-Net Classification

The model was applied to the Tully River catchment section of the aerial imagery that was excluded from the training. It took approximately two hours to create the output rasters and an additional 2.5 h to threshold, filter, and merge the classification tiles to a single polygon feature class. The result was the model-inferred extent of banana plantations, with a probability of 90% or greater.

Figure 6 shows examples of the U-Net classification of banana plantations in the Tully River catchment. In all examples the U-Net classification matches the validation data except in Figure 6(4c) where the U-Net classification was identified a new banana plantation that was missed in the validation dataset because of the young age of the plants and low canopy cover resulting in this area not being identified as a banana plantation by the human operator.

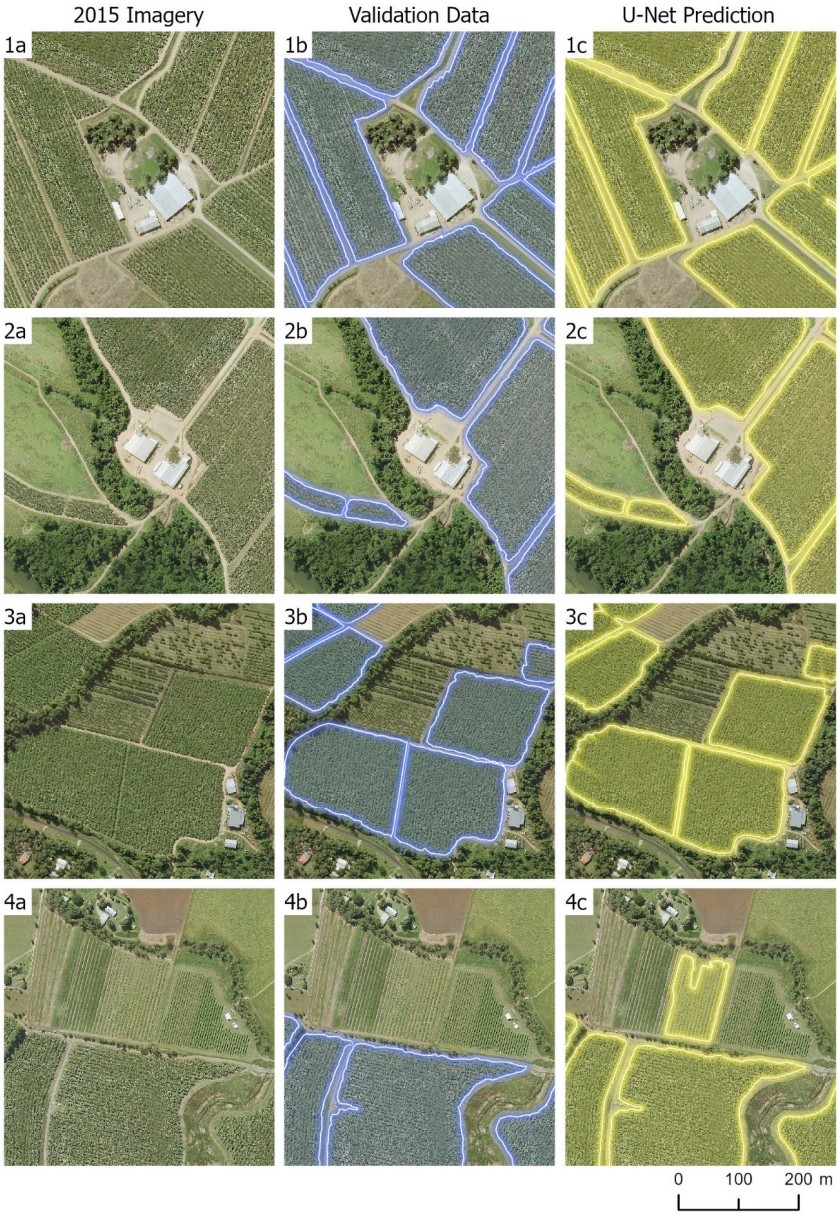

**Figure 6.** Examples (1–4) show correctly classified areas of banana plantations for the U-Net model. The left column (**a**) shows the 3-band true-color aerial photography. The middle column (**b**) shows the validation data and the right column (**c**) shows the output U-Net model classification. Note in Figure 6(4c), the U-Net classification identified a new plantation that was missed in the validation dataset.

Figure 7 shows areas where the U-Net classification did not perform well. Figure 7(1,2) show a papaya plantation and sugarcane crop respectively which the U-Net has incorrectly classified both as banana plantations. Figure 7(3) shows an area of misclassification possibly related to the effects of shadow on the banana plantation, and Figure 7(4) shows the U-Net classification not correctly classifying blurred areas of the aerial photography.

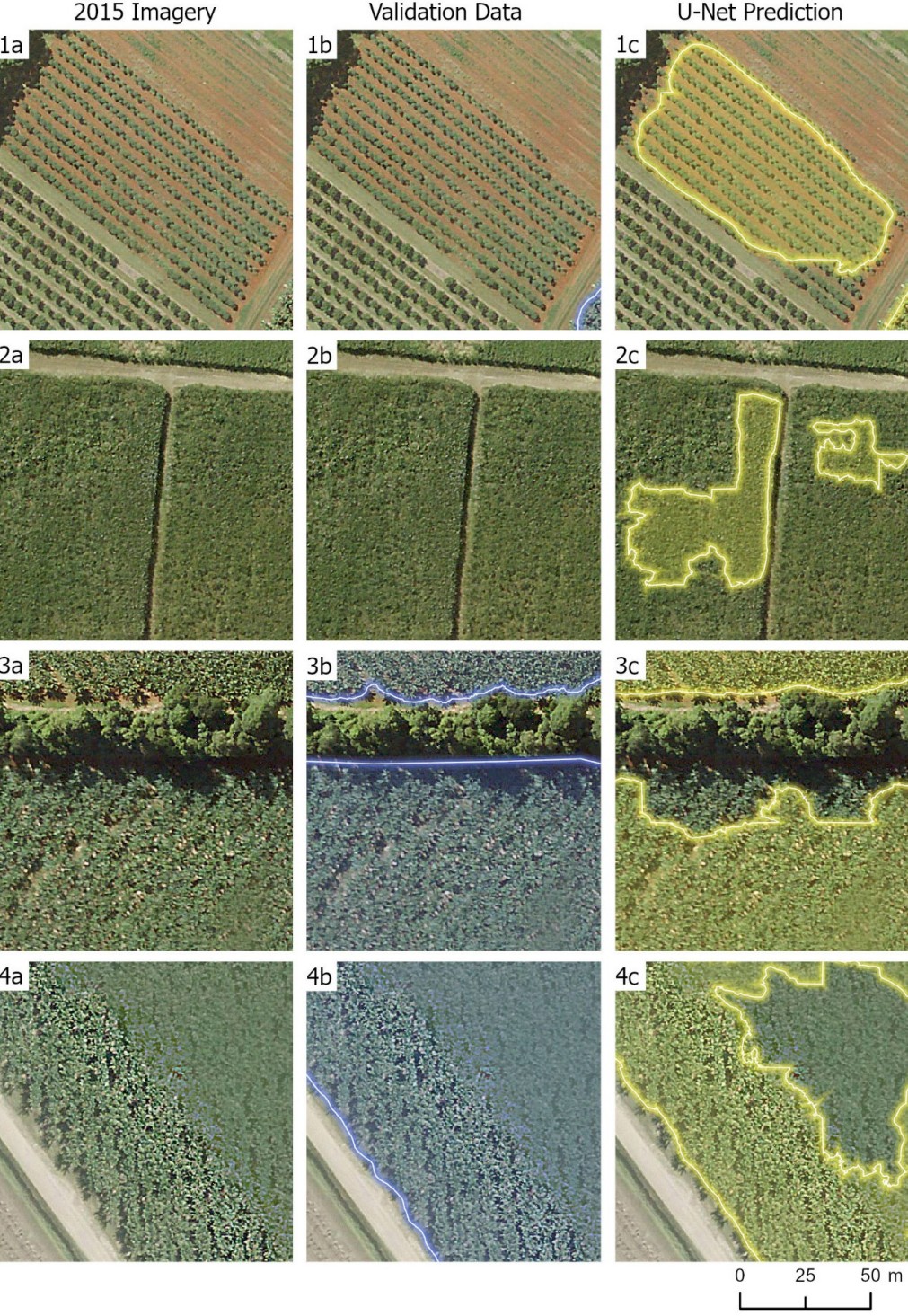

**Figure 7.** Examples (1–4) show incorrectly classified areas of banana plantations for the U-Net classification. The left column (**a**) shows the three-band true-color aerial photography. The middle column (**b**) shows the validation data and the right column (**c**) shows the output U-Net classification.

### 3.3. Accuracy Results

The results from the accuracy assessments can be found in Table 1. Overall, both classifications had a high total accuracy of >0.995. As the focus of this work is specifically mapping banana plantations which only represent 2.3% of the total area of the Tully River catchment, Table 1 shows the user's and producer's accuracy for this class. We found the QLUMP banana plantation classification had a user's and producer's accuracy of 0.862 and 0.921 respectively and the U-Net banana plantation classification had a user's and producer's accuracy of 0.983 and 0.959 respectively (Table 1). The Jaccard Index for the QLUMP classification was 0.341 and U-Net classification was 0.943.

**Table 1.** Accuracy assessment results showing the Jaccard Index, User's, Producer's, and total accuracies, assessed on independent validation data. The Total, User's, and Producer's accuracies were based on a stratified random sample of 9805 points, 701 of which were banana plantation points. The Jaccard Index used all the validation data for the Tully River catchment.

| Classification | Total Accuracy | User's Accuracy | Producer's Accuracy | Jaccard Index |
|---|---|---|---|---|
| QLUMP Banana Plantations [1] | 0.996 | 0.862 | 0.921 | 0.341 |
| U-Net Banana Plantations | 0.999 | 0.983 | 0.959 | 0.943 |

[1] The cartographic scale of the QLUMP mapping is smaller than the scale of the U-Net classification.

As Table 1 indicates that both classifications are more likely to miss areas of banana plantations (false negatives) than classify other land use features as banana plantations (false positives). It must be reiterated that the QLUMP data were compiled using lower resolution imagery and mapped land use features at a scale of 1:50,000. As a result, small-area and narrow land uses (e.g., farm sheds and roads) were aggregated into surrounding land uses. Also, banana plantations were mapped as bananas, regardless of them being active or fallow in the imagery. The scale of QLUMP data and fallow plantations affected the accuracy of the mapping, when compared with the U-Net classification. When specifically analyzing the location and extent of banana plantations, the accuracy assessment results suggest the U-Net classification is more accurate than the QLUMP data.

The quality of the imagery for U-Net is restricted to data processed by a commercial third-party. Inconsistencies in the geometric and radiometric corrections and post-processing operations (such as tile mosaicing and color balancing) have affected the output classification.

## 4. Discussion

In this paper we have presented an automated approach to mapping banana plantations using the U-Net CNN architecture [33] to assist the biosecurity response to Foc TR4. The U-Net has been successfully applied to other land uses around the world, but not at an operational level [40,54,55]. Until this study, there were no existing automated classifications to detect banana plantations using high-resolution aerial photography in Queensland, Australia or globally.

One perceived benefit of manually mapping is that humans can draw on undefined experiences or obscure learnings, such as past, present, or regional knowledge about land uses [15]. Despite this, when comparing the CNN approach to existing methods, we found the new classification technique is more accurate (>0.94). This is consistent with other similar studies mapping land uses using deep learning [34,40,54] except in this study we have applied our model over a broad geographical area.

We have also found the CNN method to be more rapid, allowing the automatic classification of banana plantations within hours when compared with existing methods requiring weeks of manually digitizing features. There were some misclassifications associated with blurred sections of aerial imagery due to third party post-processing, which caused some sections of banana plantations to be missed. Additionally, some papaya plantations and small areas of sugarcane crops were misclassified

as banana plantations. To address these issues and to create a more robust model, we suggest additional training data be generated in these problem areas to allow the CNN to better learn these features. It is important for a model classifying land use features within high resolution aerial photography to have the ability to account for these types of artefacts as past and future imagery captures are likely to contain similar issues. Stratifying the random generation of training data by land uses or targeting similar land uses (such as papaya plantations) would also ensure the CNN has enough examples of non-banana plantations, and will improve the classification result.

While the U-Net has enabled greater accuracy and a more rapid land use product, it is limited by the availability of extensive training data which is a prerequisite. Initially this work was made possible using the data produced by the QLUMP methodology which was used to map the initial extent of banana plantations in 2015. This will also be the case for any future land use classes mapped using the U-Net. Therefore, there are implications for both the QLUMP and U-Net methodologies going forward. A logical option would be to iteratively update and improve the original QLUMP data, using the proposed U-Net methodology.

Future work will focus on updating the location and extent of all banana plantations within the Wet Tropics to 2018 which was only partially available for this work (Figure 1). We will also be expanding this method to other land uses and commodities, developing methods for monitoring land use change and investigating if these methods can be used to detect damaged plantations either from wilt associated with Foc TR4 or as a result of wind damage from natural disaster such as Tropical Cyclones. Developing a framework to automatically map land use features would benefit mapping programs in Australia and globally. The automated and efficient classification of land use features from high-resolution imagery will be extremely valuable in responding to current and future biosecurity incidents as well as other events requiring a rapid response such as natural disasters. This will have applications in agricultural productivity and sustainability, land use planning, natural resource condition monitoring and investment, biodiversity conservation, and improving water availability and quality [14].

## 5. Conclusions

Current methods to map land use in Queensland are based on manual image interpretation, which are time- and resource-intensive. Land use information is fundamental for informing the response to biosecurity incidents, such as the detection of Panama TR4 in the Tully River catchment. Advances in big data and imagery availability have created an opportunity to develop methods to automatically and efficiently classify land use features over large geographical areas. This allows for higher spatial and temporal resolutions, and a more detailed classification for commodity-level observations.

In this paper we have presented an automated and efficient classification technique for detecting the location and extent of banana plantations in the Wet Tropics, which we have shown is an improvement on the existing mapping methodology. The new classification approach used a refined version of the existing QLUMP mapping to train a CNN using the U-Net architecture [33].

**Author Contributions:** Conceptualization, A.C.; methodology, A.C.; software, A.C.; validation, A.C.; formal analysis, A.C.; investigation, A.C. and J.M.; data curation, A.C.; writing—original draft preparation, A.C. and J.M.; writing—review and editing, A.C. and J.M.; visualization, A.C. and J.M.; supervision, A.C.; project administration, A.C. and J.M. All authors have read and agreed to the published version of the manuscript.

**Funding:** This research received no external funding.

**Acknowledgments:** The authors would like to thank Grant Moule for introducing us to the U-Net architecture. The authors acknowledge Robert Denham (Joint Remote Sensing Research Program) who contributed to this research by providing advice on the accuracy assessment. The authors also appreciate the reviewers' constructive feedback and suggestions, namely Matthew Pringle (Queensland Department of Environment and Science), Stuart Phinn (The University of Queensland), and Peter Scarth (Joint Remote Sensing Research Program). The authors would also like to acknowledge the Queensland Government Department of Environment and Science for providing access to the HPC infrastructure and supporting this research.

**Conflicts of Interest:** The authors declare no conflict of interest.

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
