# Peer review of "Detecting Banana Plantations in the Wet Tropics, Australia, Using Aerial Photography and U-Net"

_applsci, doi:10.3390/app10062017_

Round 1

Reviewer 1 Report

The paper tests the applicability of a U-Net neural network to map Banana Plantations in Queensland. The paper is well structured and well written.

The paper doesn't have any major issues, but has some areas where additional information would help the reader.

Issues:

  1. The paper relies heavily on the QLUMP data set. The QLUMP data set is referred to at several points in the manuscript, but key information is currently missing from the manuscript. I think the QLUMP data set should be added as a data set in Section 2 (Materials and Methods). The section should cover how the data set is created (there are references in the abstract and the discussion to it being manually created, but this should be stated in the methods section), how frequently it is updated and what the classes are (required for understanding section 2.8, where tertiary classes are referred to). You don’t need to list all the classes, but the new section should say that it is hierarchical, with however many levels and should give the number of classes at the different levels. How current is the data?
  2. Discussion – the discussion needs some comment about the data hungry nature of the U-net method and the fact that it’s only possible because of previous investment in production of QLUMP. This has implications for rolling these methods out more widely and implications for the potential use of these methods in Queensland more specifically. So you also need to add some comments about the potential uses of the method. For example, it could be used for updating/refreshing existing maps consistently given the availability of suitable quality aerial photography. The comment in line 318 about ‘manual methods’ is potentially problematic as you appear to be arguing that the manually produced QLUMP product is inferior to your U-net map and you know this because of your manually produced validation data set. I understand your point, but it may be best solved by changing ‘manual methods’ to ‘existing methods’.
  1. Given the focus on Foc TR4 in the introduction it seems strange not to revisit this in the discussion. It looks the method has potential to map existing banana plantations, as long as there are sufficient data to train the method, and then to monitor going forward. There are questions about what the deep learning method would classify infected plantations as.

Phrasing issues:

Line 17: Change ‘ The method using a..’ to ‘Using a…’

Line 19: change ‘can be applied to’ to ‘has potential for’

Line 26: The second sentence is very long and would be better split into 2 sentences.

Line 81: rephrase – at the moment it sounds like we only know that neural networks have been around for many decades because of the work by Schmidhuber. I don’t think this is quite the case, so I’d rephrase it as ‘Neural networks have been around for many decades (see review by Schmidhuber [22]’

Line 88: Change ‘Deep learning is an algorithm’ to ‘Deep learning methods are’

Line 101: add the missing ref

Line 116: I would start this paragraph, with something along the lines of ‘Previous work in Queensland, has focussed on land cover, with Pringle …’, because on my first reading I thought you were making a point about spatial resolution.

Figure 1: add a scale-bar to b)

Line 148: changed from ‘areas blurred’ to ‘blurred areas’

Line 148: delete ‘there is’

Line 219: change to ‘Sun et  al, [37] found that…’

Figures 5 & 6:check the headings at the top of the figures, should ‘Training data’ actually be ‘validation data’?

Figure 5: did the validation data miss the plantation, because it’s new? The canopy cover looks relatively low in the image.

Line 288: Delete the sentence after >0.995. Add  new sentence in to say, because the focus in this work is one the banana plantations which only cover 2.3% of the area Table 1 shows the user’s and producer’s accuracy for those classes.

Table 1: add a note in to state that the values are for the 701 banana plantation points.

Reviewer 2 Report

Dear Authors,

The article titled „Detecting Banana Plantations in the Wet Tropics, Australia, Using Aerial Photography and U-Net” describe the used the land mapping to training the U-Net neural network to detect banana plantations in the Wet Tropics of Queensland, Australia. Te methodology seems interesting, but was describe very shortly.

Line 19-20 -The method is reliable and can be applied to other commodities and land uses. – I will not agree with this statement, because this method has not been tested on other surfaces, machine learning gives different results depending on the terrain and the input material we will use.

Lines 101-103 – I lack information on what data set / and area / type of vegetation this network was trained here - Ma et al.

Line 105 - Known issues …-  this is not known to the potential recipient

Line 147-148-The quality of the imagery – were the images suitable for analysis at all?

Line 149 – if we don't have metadata then what we know about images?!

Results - too poor analysis of results, especially in the context of errors that were observed after training the network.

If the discussion describes the results in the context of other forms - land use, then it would be worth quoting the results of machine learning in such areas or also give similar problems faced by other scientists. There are too many generals information in the discussion and summary that describe only that the method has the potential, and this is obvious, but this should apply more to the minus and benefits of banana plantation research results.

Best regards,

Round 2

Reviewer 2 Report

Dear Authors,
The text has been correctly corrected according to the comments.
Best regards,

This manuscript is a resubmission of an earlier submission. The following is a list of the peer review reports and author responses from that submission.